# A comprehensive transcriptomic comparison of hepatocyte model systems improves selection of models for experimental use

Arif Ibrahim Ardisasmita [1,2,3,9], Imre F. Schene [1,2,3,9], Indi P. Joore[1,2,3,10], Gautam Kok[1,2,3,10], Delilah Hendriks[4], Benedetta Artegiani[5], Michal Mokry [1,6,7], Edward E. S. Nieuwenhuis[1,8] & Sabine A. Fuchs [2,3✉]

The myriad of available hepatocyte in vitro models provides researchers the possibility to select hepatocyte-like cells (HLCs) for specific research goals. However, direct comparison of hepatocyte models is currently challenging. We systematically searched the literature and compared different HLCs, but reported functions were limited to a small subset of hepatic functions. To enable a more comprehensive comparison, we developed an algorithm to compare transcriptomic data across studies that tested HLCs derived from hepatocytes, biliary cells, fibroblasts, and pluripotent stem cells, alongside primary human hepatocytes (PHHs). This revealed that no HLC covered the complete hepatic transcriptome, highlighting the importance of HLC selection. HLCs derived from hepatocytes had the highest transcriptional resemblance to PHHs regardless of the protocol, whereas the quality of fibroblasts and PSC derived HLCs varied depending on the protocol used. Finally, we developed and validated a web application (HLCompR) enabling comparison for specific pathways and addition of new HLCs. In conclusion, our comprehensive transcriptomic comparison of HLCs allows selection of HLCs for specific research questions and can guide improvements in culturing conditions.

[1] Division of Pediatric Gastroenterology, Wilhelmina Children's Hospital, University Medical Center Utrecht, 3584 EA Utrecht, the Netherlands. [2] Department of Metabolic Diseases, Wilhelmina Children's Hospital, University Medical Center Utrecht, 3584 EA Utrecht, the Netherlands. [3] Regenerative Medicine Center Utrecht, 3584 CT Utrecht, the Netherlands. [4] Hubrecht Institute, Royal Netherlands Academy of Arts and Sciences, Utrecht, the Netherlands. [5] Princess Maxima Center, 3584 CS Utrecht, the Netherlands. [6] Laboratory of Clinical Chemistry and Haematology, University Medical Center, Utrecht, the Netherlands. [7] Department of Cardiology, University Medical Center, Utrecht, the Netherlands. [8] Department of Sciences, University College Roosevelt, 4331 CB Middelburg, the Netherlands. [9] These authors contributed equally: Arif Ibrahim Ardisasmita, Imre F. Schene. [10] These authors contributed equally: Indi P. Joore, Gautam Kok. ✉email: S.Fuchs@umcutrecht.nl

To accurately study human liver physiology and pathology, in vitro models should faithfully replicate in vivo liver functions. These include elimination of toxins, production and secretion of plasma proteins and bile, and metabolic homeostasis of carbohydrates, amino acids, and lipids. Most of these processes are performed by hepatocytes, epithelial cells that constitute 60% of the number of cells and 80% of the volume of the liver[1]. As such, the pursuit of in vitro models that possess robust hepatocyte functionality remains a major goal of biotechnology.

Freshly isolated primary human hepatocytes (PHHs) represent the gold standard to investigate liver functions. However, standard two-dimensional PHH cultures are difficult to expand and rapidly lose hepatic functions[2]. To overcome these limitations, many groups have attempted to improve long-term PHH culturing methods, stimulating proliferation or minimizing dedifferentiation[3–7]. Additionally, hepatic in vitro models were established from other cell sources, including fetal hepatocytes, intrahepatic cholangiocytes, pluripotent stem cells, fibroblasts, urinary cells, and mesenchymal stem cells[4,8–56]. We here collectively designate these models 'hepatocyte-like cells' (HLCs).

The hepatic phenotype is likely to differ between different HLCs, depending on the cell of origin and culturing protocols. Clarifying these differences and identifying the best performing model is required to select the appropriate HLC model to study a specific biological or clinical question. In this study, we set out to compare hepatocyte functionality between the final stage of each HLC protocol. A systematic search of the literature and analysis of reported functional assays and expression of individual genes of the different HLCs did not allow thorough comparison of the hepatic phenotype between studies. Therefore, we developed a computational algorithm for comparison of whole transcriptome sequencing (RNA-seq) data across different studies and a web application to add additional HLC datasets in the future (HLCompR, https://github.com/iardisasmita/HLCompR). This resource will guide selection of HLCs tailored to a specific research aim and help to improve HLC culturing protocols.

## Results

**Reported hepatocyte functions are insufficient for HLC comparison.** We searched the literature for articles that described HLC culture protocols and characterized hepatic functions. As cross-study comparison of HLCs requires a universal standard, we only considered studies that tested HLCs alongside PHHs. This strategy yielded 53 studies describing HLCs derived from pluripotent stem cells (PSCs), fibroblasts, mesenchymal stem cells, urinary cells, intrahepatic cholangiocytes, and PHHs (Fig. 1a and Supplementary Fig. 1). For our quantitative comparison, we considered hepatic functional assays that were performed in more than 10 studies, the expression of associated genes, and genes commonly used as hepatic markers (Supplementary Data 1).

Some functions, including albumin secretion and CYP3A4 activity were assessed in most (>50%) studies, while other important liver functions, including bile secretion, cholesterol metabolism, and gluconeogenesis were generally left unaddressed (Fig. 1a and Supplementary Data 1). Moreover, many studies did not include PHHs in all functional assays. Similarly, only the RNA expression of *ALB*, *CYP3A4*, *CYP1A2*, and *CYP2C9* were reported in the majority (>50%) of studies while other important hepatic markers were lacking.

For cross-study comparison of liver functionality we considered functional activity of HLCs, calculated as a percentage of the PHH control included in the same study (Fig. 1a). Using this method, different hepatic functions seemed to evolve independently from each other. For example, high albumin secretion

correlated with high CYP3A4 activity in PSC-derived HLCs of Wang et al.[46], but not in PSC-derived HLCs of Boon et al.[30]. This suggests that it is impossible to predict overall hepatocyte maturation using only a single hepatic function.

Importantly, different studies did not specify the culture time of PHH controls. This may have caused significant variability in control PHH functionality, as specific hepatic functions, including CYP3A4 activity, are rapidly lost during PHH culturing[12,57] (Fig. 1b). As such, minor variations in culture duration and assay procedures of control PHHs may have profound effects on the relative activity of liver functions in HLCs.

Together, the frequent omission of PHH controls, the narrow range of functional assays performed, and the possible variability of PHH controls made it impossible to compare the hepatic phenotype between HLCs based on reported assays.

**Transcriptomic comparison reveals distinct liver-specific molecular signatures in HLCs.** To allow transcriptome-wide and standardized comparison of HLCs, we developed a computational algorithm to analyze raw bulk RNA-seq data from different HLC studies, that included PHHs or liver tissue as universal controls (Fig. 2a). This yielded 11 studies describing HLCs derived from hepatocytes, intrahepatic cholangiocytes, fibroblasts, and PSCs (Fig. 2b and Supplementary Data 2). In addition, protocols developed by Huch et al.[17] and Hu et al.[4] are commonly used in our laboratory and elsewhere to generate intrahepatic cholangiocyte-derived organoids and (fetal) hepatocyte-derived organoids, respectively. Since both studies did not provide bulk RNA-seq data, we generated RNA-seq data of HLCs derived from intrahepatic cholangiocytes (Huch-Chol-HLCs) and fetal hepatocytes (Hu-FHep-HLCs) using the corresponding protocols[4,17,58]. We did not include the adult hepatocyte-derived organoids because we were not able to culture them over long periods of time and, to the best of our knowledge, nor were other groups[58,59]. The protocols used to generate the different HLCs are represented in Supplementary Fig. 2. In addition to the HLCs and their respective PHH controls, we included fetal hepatocytes, PSCs, fibroblasts, hepatoma cell line HepG2, and common bile-duct tissue (CBD[60], i.e., extrahepatic cholangiocytes; Supplementary Data 2).

Principal component analysis (PCA) showed that samples clustered by cell type rather than by study, confirming that our computational approach allows cross-study comparison (Fig. 2c). This also confirmed transcriptional homogeneity of PHH/liver samples from different studies, which can collectively serve as a common hepatic benchmark. Most HLCs clustered closely to their respective cells of origin, with hepatocyte-derived HLCs (Hep-HLCs) clustering most closely to PHHs and liver tissue (Fig. 2c). Fibroblast-derived HLCs (Fib-HLCs) from Xie and Du clustered strikingly close to Hep-HLCs, while Fib-HLCs from Gao grouped closer to fibroblasts. Du and Gao used similar media compositions but different transcription factors (TFs) for transdifferentiation (Fig. 2b), showing better hepatic reprogramming using the TF combination of Du. The protocols differentiating intrahepatic cholangiocytes towards hepatocytes (Chol-HLCs) resulted in HLCs that clustered with CBDs. Interestingly, the PSC-derived HLCs (PSC-HLCs) from Mun, which underwent a final maturation step according to the intrahepatic cholangiocyte culture protocol of Huch et al.[17,24] (Supplementary Fig. 3), also clustered closer to CBDs than to either PSCs or PHHs. This suggests that the culture protocol of Huch et al.[17] directs differentiation towards cholangiocytes rather than hepatocytes. Considering other PSC-derived HLCs (PSC-HLCS), we observed that the HLCs from Wang, which were derived from PSCs with the ability to form extra-embryonic tissues[61], clustered closer the

**a**

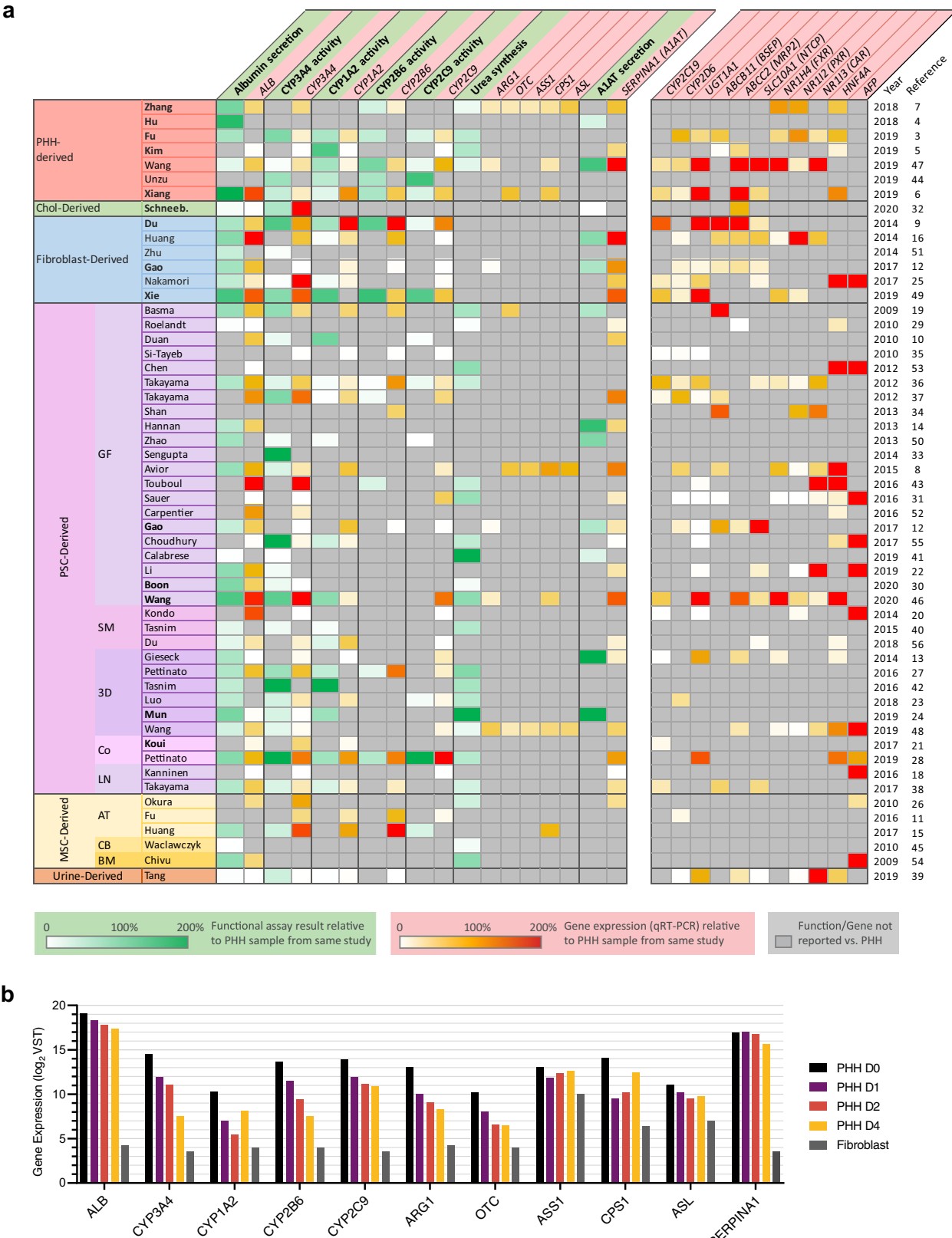

**b**

Hep-HLCs than the HLCs derived from standard PSCs (Gao, Mun, Boon, Koui; Fig. 2c).

We established a general hepatic fingerprint, by considering the top-expressed genes in the liver from the Genotype-Tissue Expression (GTEx) database. Hierarchical clustering showed that only Hep-HLCs clustered with PHHs and liver tissue based on

these top-expressed liver genes (Fig. 2d). To quantify the overall resemblance of HLCs to PHHs according to a given gene set, we calculated distance-based similarity scores (DBS) (Fig. 2e). Hep-HLCs showed the highest DBS based on top-expressed liver genes, followed by Wang-PSC-HLCs, Fib-HLCs, other PSC-HLCs, and finally Chol-HLCs (Fig. 2e).

**Fig. 1 Reported hepatic functional assays and gene expression are insufficient for HLC comparison. a** The most commonly reported hepatic functions and gene expression are presented. The value in each cell represents the activity or expression level of an HLC as a percentage of the PHH control in the same study. HLCs are grouped based on the type of cells they were generated from. Pluripotent stem cell (PSC) derived HLCs are further grouped into: GF, standard protocols employing growth factors; SM, protocols solely using synthetic molecules; 3D, protocols utilizing 3D matrices; Co-cult, protocols combining multiple cell types; and LN, protocols focusing on the effect of laminin coating. Mesenchymal stem cell (MSC) derived HLCs are categorized into the tissue of origin: AT adipose tissue, CB cord blood, BM bone marrow. **b** mRNA expression (log2) of genes relevant to the commonly reported hepatic functions in PHHs cultured for 0, 1, 2, and 4 days and cultured dermal fibroblasts, from the study of Gao et al.[12] Note that expression of some genes (e.g. *CYP3A4, CYP1A2, OTC*) decreases >4-fold within the first 24 h of culturing, whereas expression of other genes (*ALB, ASS1, SERPINA1*) remains relatively stable. Source data are provided in Supplementary Data 3.

Next, we considered several gene sets related to specific hepatic functions. HLCs derived from hepatocytes, fibroblasts from Xie, and extended PSCs from Wang were generally most similar to PHHs, but clear differences could be observed when considering particular gene sets (Fig. 2f and Supplementary Figs. 4 and 5). For example, expression of most gene sets was higher in Hep-HLCs than in Fib-HLCs, except for the gluconeogenesis gene set. Among all Hep-HLC samples, Xiang-Hep-HLC-D15 was generally most similar to PHHs and liver tissue (Fig. 2f), which may result from the relatively short culture time (15 days) of this HLC sample compared to other HLCs (>3 weeks)[3,5,6]. We also observed that Chol-HLCs remained highly similar to CBDs in overall gene expression (Fig. 2c), with a slight increase in expression of genes involved in cholesterol metabolism and bile secretion (Fig. 2f). In HepG2 cells, specific gene sets including cholesterol and glycogen metabolism and complement production were relatively well expressed.

We then evaluated whether the expression of gene sets related to specific liver functions reflected differences in zonation between HLCs. To this end, we extracted periportal and pericentral gene modules from the single-cell analysis of human hepatocytes performed by Aizarani and colleagues[62]. To ensure that these modules assessed hepatic specific genes associated with hepatocyte zonation, we only included zonation genes that were enriched (>2-fold) in PHH/liver samples compared to other HLC cell sources (Fib, PSC, Chol) (Supplementary Fig. 6). This approach revealed that most HLCs expressed periportal and pericentral modules in a linear manner (Fig. 2g). This may suggest that either there are no strong zonation patterns in most HLCs (assuming homogenous gene expression profiles in all cells) or there are zonation patterns that cannot be discerned due to the nature of bulk sequencing analysis. Single-cell RNA sequencing analysis is needed to fully address zonation in HLCs. Interestingly, Chol-HLCs and HepG2 cells deviated most from this general linear pattern. Chol-HLCs displayed a predominant periportal identity (Fig. 2g) which corresponded to relatively high expression of gluconeogenic genes (Fig. 2f). In contrast, HepG2 cells exhibited a more pericentral identity (Fig. 2g), corresponding to higher expression of genes involved in cholesterol metabolism[63] (Fig. 2f).

By considering HLCs from all available studies, our approach also revealed the relative magnitude of hepatic differences between HLCs generated using various protocols within individual studies. For instance, the HLCs in the study of Gao were either directly transdifferentiated from fibroblasts (Fib-HLCs) or generated through an intermediate iPSC step (PSC-HLCs). Gao et al.[12] observed that their Fib-HLCs performed better at Phase I and II reactions, whereas their PSC-HLCs modeled hepatic fatty acid metabolism better. Our analysis confirmed these findings (Fig. 2f and Supplementary Figs. 4e and 5a) but also demonstrated that both the Fib-HLCs and PSC-HLCs from the Gao study displayed relatively weak hepatic phenotypes compared to HLCs from other studies. Furthermore, Boon et al.[30] attempted to enhance the maturation of PSC-derived HLCs with TF transduction (*HNF1A, FOXA3, PROX1*) and showed that this approach

resulted in higher albumin secretion and CYP3A4 activity. However, in relation to other HLCs, TF transduction only resulted in a slight improvement of hepatocyte differentiation (Fig. 2c, f, g). Moreover, TF transduction not only increased (e.g., *ALB* and *HPX*) but also decreased (e.g., *APOA2* and *TTR*) the expression of some specific liver markers in the PSC-HLCs of Boon et al.[30] (Fig. 2d).

**HLCs display different cell/tissue identities**. The quality of hepatocyte in vitro models should not only be defined by the presence of hepatocyte identity (Fig. 2) but also by the absence of unwanted cell/tissue identities. We therefore analyzed the HLCs using CellNet[64,65], a platform that quantifies resemblance to a larger variety of human cell and tissue types (cell/tissue classification score) based on establishment of tissue-specific gene regulatory networks (GRN status).

CellNet classified most HLCs as liver but only the (fetal) hepatocyte-derived HLCs attained a pure liver classification. The HLCs derived from PSCs, fibroblasts, and cholangiocytes were either classified as multiple cell/tissue types or non-liver tissues (Fig. 3a). The unwanted cell identities of these HLCs can be attributed to incomplete loss of their original cell identity or undesired gain of non-liver identity.

The incomplete loss of original cell identity could be observed in all PSC-HLCs, which showed a higher embryonic/pluripotent stem cell (ESC) GRN status compared to PHHs (Fig. 3b and Supplementary Fig. 7). Interestingly, despite retaining discernible ESC GRN status, the PSC HLCs from Wang displayed similar liver GRN status to Hep-HLCs (Fig. 3b). For Fib-HLCs, only the samples from Gao failed to fully extinguish their fibroblast GRN status (Fig. 3b and Supplementary Fig. 7). This incomplete loss of fibroblast GRN and partial gain of liver GRN in Fib-HLCs from Gao resulted in low classification scores for all specific cell/tissue types (Fig. 3a). This heterogeneity in cell identity between HLCs starting from the same cell source (Wang-PSC-HLCs vs. other PSC-HLCs and Gao-Fib-HLCs vs. other Fib-HLCs) reflects the effects of different culture protocols applied in each study.

All non-hepatocyte-derived HLCs manifested undesired gain of intestine/colon identity (Fig. 3a). Additionally, PSC-HLCs from Koui also gained lung identity possibly due to their protocol involving differentiation towards multiple lineages (Fig. 3a and Supplementary Fig. 2). This gain of non-liver identity may occur in all cells or only in a subpopulation of cells, resulting from heterogeneous differentiation. Regardless, this suggests that all HLC generation protocols are still imperfect and fine-tuning the cell fate specification of these HLCs may improve hepatic (trans-)differentiation[66]. Surprisingly, the Chol-HLCs from Huch and Schneeberger and PSC-HLCs from Mun bore higher resemblance to the intestine/colon than to the liver (Fig. 3a). Furthermore, our CBD control samples had lower intestine/colon classification than the Huch protocol-cultured HLCs (Fig. 3a). Nevertheless, PCA-analysis showed that the Huch protocol-cultured HLCs were more similar to CBD than to intestine/colon samples (Supplementary Fig. 3b), suggesting that the Huch protocol not only promotes CBD but also intestinal gene expression. This is in line

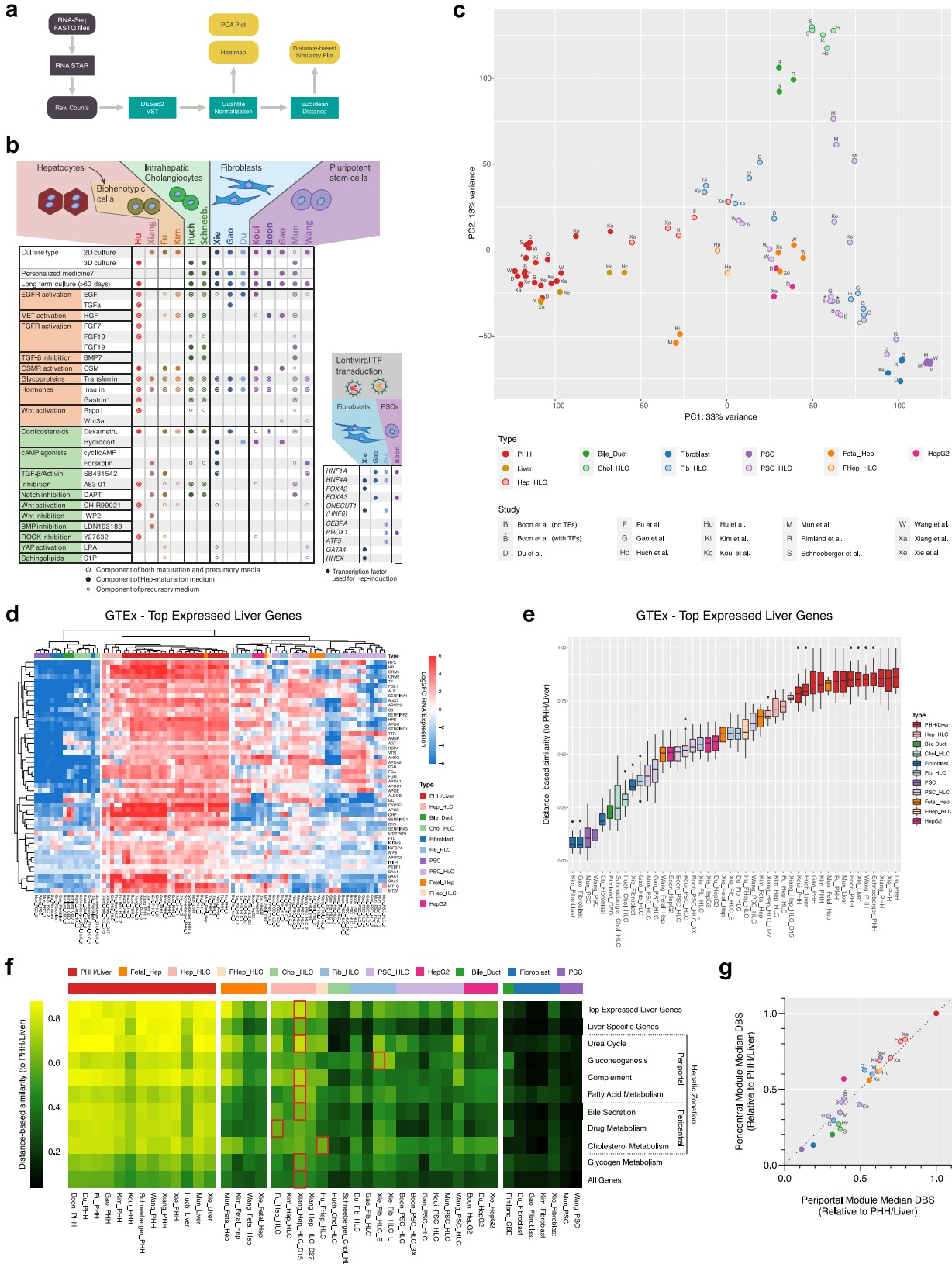

with the finding of Aizarani et al. that intrahepatic cholangiocytes upregulates intestinal marker genes when cultured as organoids in Huch expansion medium[62].

Besides unwanted adult cell/tissue identities, HLCs may also display immature hepatic identity, resembling fetal hepatocytes[67]. Since CellNet was trained to only distinguish between adult tissue identities, we created a classification algorithm based on the transcriptome of adult and fetal hepatocytes. Corresponding with the CellNet results, the Hep-HLCs of Fu and Xiang best resembled the adult hepatocyte transcriptome (Fig. 3a, c). Interestingly, none of the HLCs displayed fully mature hepatocyte fingerprints, resulting from absence of adult markers or presence

**Fig. 2 Transcriptomic comparison reveals distinct liver-specific molecular signatures in HLCs. a** Schematic of the algorithm employed to conduct the transcriptomic comparison analysis. All steps in gray were performed on the Galaxy web platform. **b** Summary of protocols used to generate HLCs that were included in the transcriptomic comparison analysis. Precursory medium is any medium that was used directly prior to the final maturation medium, including progenitor/expansion medium (hepatocyte-, cholangiocyte-, and fibroblast-derived HLCs) and hepatocyte differentiation medium (PSC-derived HLCs). **c** Principal component analysis created using 5000 genes with the highest variance among all samples from different studies. **d** Heatmap of top-expressed liver genes according to the Genotype-Tissue Expression (GTEx) project. Hierarchical clustering was performed using Euclidean distance.
**e** Distance-based similarity score (DBS) calculated using Euclidean distance describing the resemblance between each sample to all PHH/Liver samples based on the top-expressed liver gene set. Box-and-whisker plots are shown as median (line), interquartile range (box), and data range or 1.5x interquartile range (whisker). **f** Heatmap showing the median DBS of all samples using various liver function associated gene sets. Red outlines indicate the HLC with the highest DBS for each gene set. **g** The median DBS of all samples relative to PHH/Liver based on the pericentral and periportal modules (see Fig. 2c for legend). Source data are provided in Supplementary Data 4 and 5.

of fetal markers (Fig. 3c). The Chol-HLCs were not classified as either adult or fetal hepatocytes (Fig. 3c), underlining their weak hepatic differentiation. Among all fibroblast- and PSC-derived HLCs, only the Fib-HLCs of Xie attained a strong adult identity, expressing mature hepatocyte markers including *ADH1B* and *SRD5A2*. Despite convincing liver GRN status in the CellNet analysis, the PSC-derived HLCs from Wang still demonstrated a fetal identity (Fig. 3b, c).

For HLC protocols using TF transduction, the extent of hepatic (trans-) differentiation is determined in part by the effectiveness of the TF combinations used. This effectiveness can be assessed using the network influence scores from CellNet analysis. Network influence scores indicated that TF transduction in Fib-HLCs of Gao (*HNF4A, FOXA3, HNF1A*) and PSC-HLCs of Boon (*PROX1, FOXA3, HNF1A*) increased TF expression and activation of associated TF networks, without resulting in activation of other liver TF networks (Fig. 3d). In contrast, the additional TFs used in the Fib-HLCs of Du (*ATF5, PROX1, CEBPA, ONECUT1* (also known as *HNF6*)) and Xie (*ONECUT1, FOXA2, GATA4*) yielded widespread activation of other liver TF networks. Of note, low RNA expression and low network influence scores suggested ineffective transduction of *ATF5* and *PROX1* in one of the Du samples (Du-Fib-HLC-1), but effective transduction of *ONE-CUT1*, which still resulted in broad hepatic TF network activation. Together, this suggests that *ONECUT1* is an important factor in the hepatic transdifferentiation cocktail.

Additionally, we used CellNet to validate our computational algorithm and found that the PHH/liver, fibroblast, and PSC samples included in our analysis showed high similarity to their corresponding CellNet training datasets (Fig. 3a, b). Furthermore, both the ranked CellNet liver classification score and liver GRN status correlated well with the DBS score (using all genes) obtained through our own approach (Fig. 3e, f). This confirmed that our transcriptomic analysis accurately quantified liver resemblance.

**Transcriptomic comparison allows prediction of HLC functionality**. Our transcriptomic comparison indicated substantial variability in the expression of liver-related gene sets between included HLCs (Fig. 2f). To determine if our comparison could predict the ability of specific HLCs to model liver functions, we assessed whether transcriptomic differences translate into proteomic and ultimately functional differences.

We first considered publicly available proteomes of HepG2 cells and PHHs[68] and found that the transcriptomes of HepG2 cells and PHHs from our dataset correlated very well ($R^2 = 0.40$, $p < 0.0001$; Fig. 4a) to the proteomes from this independent study[68]. In fact, this correlation was comparable to within-study transcriptome-proteome correlations[69] ($R^2 = 0.41$). Transcriptomic and proteomic enrichment analyses concurrently indicated that HepG2 cells display higher expression of cell cycle pathways and lower expression of liver-related pathways, compared to PHHs (Fig. 4b).

We next assessed if transcriptional differences translate into functional differences by considering Hepatitis B virus (HBV) infection. This disease has been successfully modeled using Hep-HLCs from Xiang and Fib-HLC form Xie[6,49]. Accordingly, the expression of the genes associated with HBV infection and propagation was best recapitulated by these two HLCs (Fig. 4c). This confirms that for HBV, our transcriptomic comparison could predict suitability for in vitro modeling.

We further validated the predictive power of our transcriptomic comparison by considering reported albumin secretion (Fig. 1a) because the gene expression of *ALB* remains stable for several days in cultured PHHs (Fig. 1b). As such, the relative albumin secretion of different HLCs is reasonably unaffected by the variability of the PHH controls, and can therefore be compared across studies. The expression of *ALB* in our transcriptomic comparison was a good predictor ($R^2 = 0.69$; $p = 0.001$) of reported albumin secretion for the 11 included HLCs (Fig. 4d).

Finally, we validated that our transcriptomic comparison can also predict the activity of functions that are highly affected by the freshness of the internal PHH controls, such as CYP3A4 and urea cycle activity (Fig. 1b). To ensure comparability, we performed assays of these functions in a uniform setup using HLCs that are well-established in our group (Chol-HLCs, FHep-HLCs, and HepG2 cells). We found that the enzymatic activity of CYP3A4 was highest in Chol-HLCs (Fig. 4e), which was correctly predicted by the transcriptomic comparison (Fig. 4f). Considering urea cycle gene expression, FHep-HLCs resembled PHH/Liver most, followed by HepG2 cells and Chol-HLCs (Fig. 4g). In line with this, FHep-HLCs displayed highest urea secretion among these HLCs (Fig. 4h). The smaller expressional difference between Chol-HLCs and HepG2 cells (Fig. 4g) did not result in differences in urea secretion levels (Fig. 4h), possibly because two essential enzymes of the urea cycle (*CPS1* and *OTC*) are lowly expressed in both Chol-HLCs and HepG2 cells (Fig. 4i). In fact, both HLCs had expression levels similar to fibroblasts (Fig. 4g, i), which also poorly convert ammonia to urea[70]. Together, these findings supported that our cross-study transcriptomic comparison predicts HLC protein abundance and functionality.

**HLCompR web application for HLC selection**. To allow other researchers to examine the expression of an important gene or an entire gene set to select appropriate HLC models, we created HLCompR, a web application for easy exploration of the relative expression of any gene (set) of interest (https://github.com/iardisasmita/HLCompR). Additionally, HLCompR allows researchers to filter HLCs by specific characteristics, including culture duration, expandability, and cell source, and easily select function- or disease- associated gene sets (Fig. 5a).

To illustrate how HLCompR can help to select the optimal HLCs for specific research purposes, we considered liver diseases

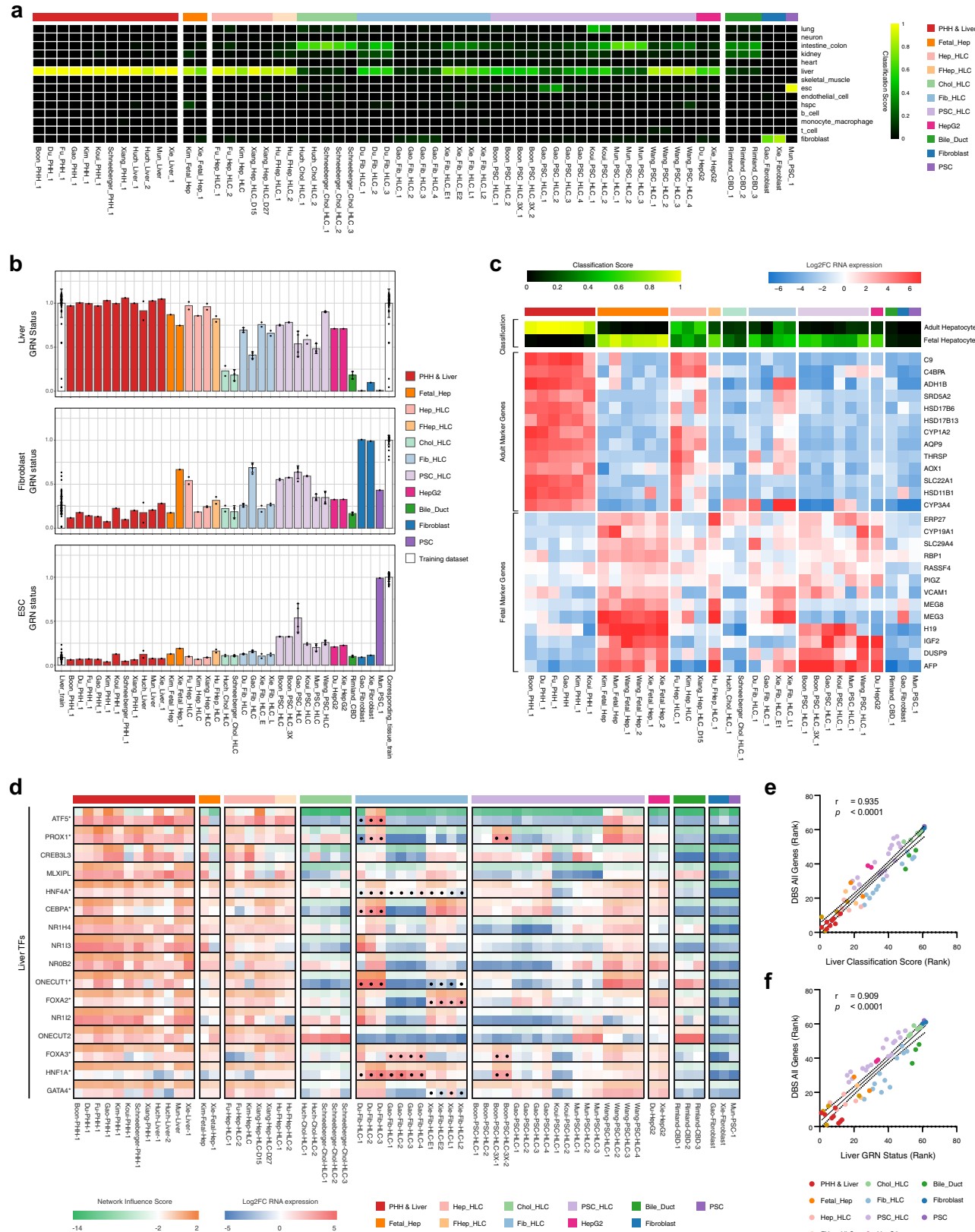

associated with a single gene defect (Fig. 5b). This showed that multiple HLCs express the *SERPINA1* gene on a similar level to PHHs, and might therefore be appropriate to study protein secretion and activity in alpha-1-antitrypsin deficiency. In contrast, only a few HLCs may optimally model the function of the enzyme affected in glycogen storage disease type 1a (*G6PC*;

Fig. 5b). The optimal choice for modeling this disease would be the Fib-HLCs from Xie.

For more complex liver diseases associated with multiple genes, the distance-based similarity score of the disease-associated gene set provides a better overview of the modeling capabilities of various HLCs. For example, Hep-HLCs better resembled the

**Fig. 3 HLCs display different cell/tissue identities. a** Cell/tissue classification heatmap showing different cell/tissue identities of various HLCs. The classification scores represent the probability that the samples express indistinguishable gene regulatory networks (GRN) from the training dataset. **b** Liver, fibroblast, and embryonic stem cell (ESC) GRN status of HLCs and CellNet training datasets expressed as mean values. Each dot and error bar represents an individual sample and the standard deviation, respectively. **c** Heatmaps showing the adult/fetal hepatocyte classification score and expression of representative marker genes for the adult and fetal hepatocyte identities. **d** Heatmap showing the network influence scores (NIS) and expression values of liver-associated transcription factors (TFs). Asterisks and dots indicate the TFs used for transductions. **e** Spearman correlation between the rank of CellNet liver cell/tissue classification score or **f** CellNet liver GRN status and the rank of the DBS for all genes. Lower ranks represent higher liver classification, GRN status, and DBS. Spearman's correlation coefficient (*r*) and *P* value (*p*) are shown on the top left of the graphs (*n* = 62 biologically independent samples). Dotted lines represent the 95% confidence interval. Source data are provided in Supplementary Data 6.

hepatic expression of genes related to non-alcoholic fatty liver disease (NAFLD) compared to HepG2 cells, which are commonly used to model NAFLD[71] (Fig. 5c). In addition to transcriptional resemblance to PHH, NAFLD modeling requires long-term culturing of fully matured HLCs. Therefore, Hep-HLCs from Xiang, which can be maintained in a differentiated state for 1 month, may represent the optimal in vitro model for this disease.

**Addition of new HLC transcriptomes to the HLCompR web application**. Given the technical challenges and variability inherent to HLC protocols, replication of such protocols might not yield phenotypically identical HLCs. In addition, new HLC protocols are continually developed. We therefore supported addition of new RNA-seq data to the HLCompR analysis using read counts processed according to our pipeline as inputs (https://github.com/iardisasmita/HLCompR).

To determine whether new datasets can be compared using HLCompR, we tested publicly available RNA-seq datasets of PHH or liver tissue samples in the application[48,72–76]. We also included several studies that did not meet our RNA-seq inclusion criteria[43,44] (query dataset) (Supplementary Fig. 1). The samples included in Fig. 2 served as the training dataset for HLCompR.

The query dataset from Gupta et al.[74] was compatible with HLCompR because their PHH and liver samples clustered together with the training samples (Fig. 6a). Accordingly, these samples acquired high liver classification and liver GRN status scores in CellNet (Fig. 6b, c). Query datasets of Shi et al.[75] and Vieyres et al.[76] were also compatible with HLCompR, even though their PHH clustered in between the training PHH/liver samples and hepatocyte-derived HLCs (Fig. 6a and Supplementary Fig. 8a). CellNet classified their PHH samples as liver, but their liver GRN status was slightly lower than training PHH/liver samples (Fig. 6b, c and Supplementary Fig. 8b, c).

The query dataset from Unzu et al.[44] was incompatible with HLCompR, because the PHHs from their dataset did not cluster with the training data, but the PSCs did (Fig. 6a). Concurrently, analysis with CellNet showed incorrect classification of PHHs and correct classification of PSCs (Fig. 6b, c). Similarly, PHHs included in the query datasets from ter Braak et al.[72], Wang et al.[48], and Guan et al.[73] did not cluster with the training dataset (Fig. 6a and Supplementary Fig. 8a), which was also reflected by low CellNet liver scores (Fig. 6b, c and Supplementary Fig. 8b, c). Samples from Touboul et al.[43] clustered close to the respective training samples on principal component 1 (PC1) but were separated by principal component 2 (PC2) (Fig. 6a). Still, CellNet analysis showed classification scores similar to those of training samples (Fig. 6b, c). Interestingly, when changing the number of genes considered for PCA or reducing the number of samples included (Supplementary Fig. 8d), the samples of Touboul et al.[43] clustered together with the training samples. Nevertheless, we considered the samples from Touboul et al.[43] incompatible with HLCompR. We recommend use of HLCompR only when control PHH/liver samples are comparable to the training PHH/liver

samples. We added a Random Forest classifier that automatically reports compatibility of a new dataset based on this parameter.

Since standardized mapping and RNA-seq processing may still result in incompatibility with HLCompR, we hypothesized that the type of RNA-seq library preparation or sequencer might influence compatibility (Supplementary Data 2). Based on this possibility, we suggest that libraries should be prepared using standard Illumina TruSeq RNA sample preparation kit and sequenced with Illumina sequencing machines. As we cannot guarantee compatibility, new datasets should always include PHH/liver control samples.

When a dataset is compatible with HLCompR, new HLC samples can be compared to the other HLCs in the training dataset. For example, in the query dataset from Gupta et al.[74], the hepatically differentiated hepatoma cell line HepaRG showed better resemblance to PHHs than HepG2 cells (Fig. 6a). Correspondingly, HepaRG cells have also been reported to perform better at hepatic functional assays than HepG2 cells[77].

## Discussion

The availability of myriad hepatocyte model systems gives researchers the option to select one that best suits their study, but an educated choice is hampered by the absence of standardized evaluation. We addressed this by performing a comprehensive cross-study comparison of HLCs, derived from various cell sources. Our literature search showed that only a small subset of hepatic functions is routinely tested in direct comparison to PHH controls, precluding cross-study comparison of HLCs. Furthermore, as cultured PHHs rapidly lose specific hepatic functions, a major determinant of the relative activity of these functions in HLCs is the culture time of control PHHs. Since various HLC studies use control PHHs cultured for different durations, the relative activities of most reported hepatic functions cannot be reliably compared across studies. Therefore, we not only recommend that protocols describing new HLCs report a set of minimal hepatic characteristics[78], but we also stress the importance of using standardized assays and inclusion of control PHHs that are cultured for a standardized duration.

Using currently available RNA-seq data, we developed a computational algorithm for in-depth cross-study comparison of HLCs with minimal study-specific batch effects. This allowed evaluation of genes that are rarely investigated in individual studies but may be important for specific hepatic functions or liver disease models. Our analysis revealed that the transcriptomic profile of HLCs is determined by the cell of origin and the protocol used, with hepatocyte-derived HLCs most closely resembling PHHs. In addition, we identified hepatic marker genes that are lowly expressed in most HLCs, including *CYP2E1*, *ADH1A*, *F9*, and *SERPINC1*. These genes may serve as important indicators of mature hepatic differentiation, besides common markers such as *ALB*, *CYP3A4*, and *SERPINA1*.

We found that hepatic genes are differentially expressed between different HLCs. This underlines the importance of selecting the most appropriate HLC for a specific research

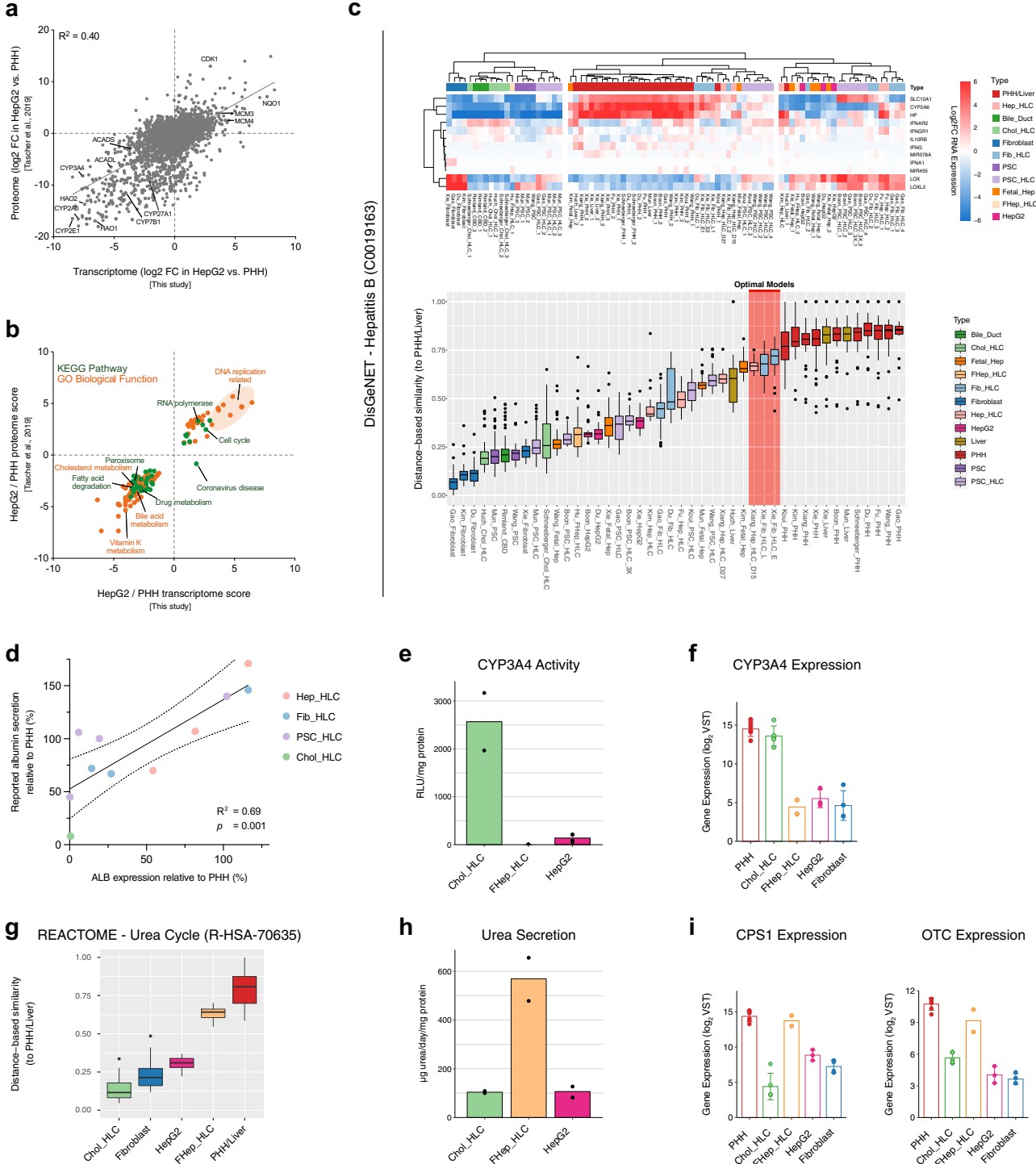

**Fig. 4 Transcriptomic comparison allows prediction of HLC functionality. a** Scatter plot of protein abundance ratios against corresponding mRNA ratios in HepG2 cells vs. PHHs for proteins/genes that are detected in both transcriptome (from our transcriptomic comparison) and proteome (from Tascher et al.[68]). Pearson's coefficient of determination ($R^2$) is shown at the top left of the plot. **b** Scatter plot of enriched pathways based on transcriptome and proteome. Only pathways that are enriched on both the transcriptomic and proteomic levels are shown. **c** Heatmap and DBS of hepatitis B associated genes according to the DisGeNET database. Red highlight indicates the predicted optimal hepatocyte in vitro models. **d** Correlation plot of the gene expression levels of ALB from the transcriptomic comparison data (x-axis) and the reported relative albumin secretion (y-axis) in included HLCs. Pearson's coefficient of determination ($R^2$) and P value (p) are shown on the bottom right of the graph ($n = 11$ biologically independent samples). Dotted lines represent the 95% confidence interval. **e** CYP3A4 activities presented as mean. **f** Gene expression level of *CYP3A4* from the transcriptomic comparison data presented as mean ± standard deviation. **g** The DBS of urea cycle genes according to the Reactome database. The combined values of Chol-HLC, fibroblast, HepG2 cell, FHep-HLC, and PHH/liver samples from different studies are presented. **h** Urea secretion level presented as mean. **i** Gene expression level of *CPS1* and *OTC* from the transcriptomic comparison data presented as mean ± standard deviation. Box-and-whisker plots are shown as median (line), interquartile range (box), and data range or 1.5x interquartile range (whisker). Source data are provided in Supplementary Data 7.

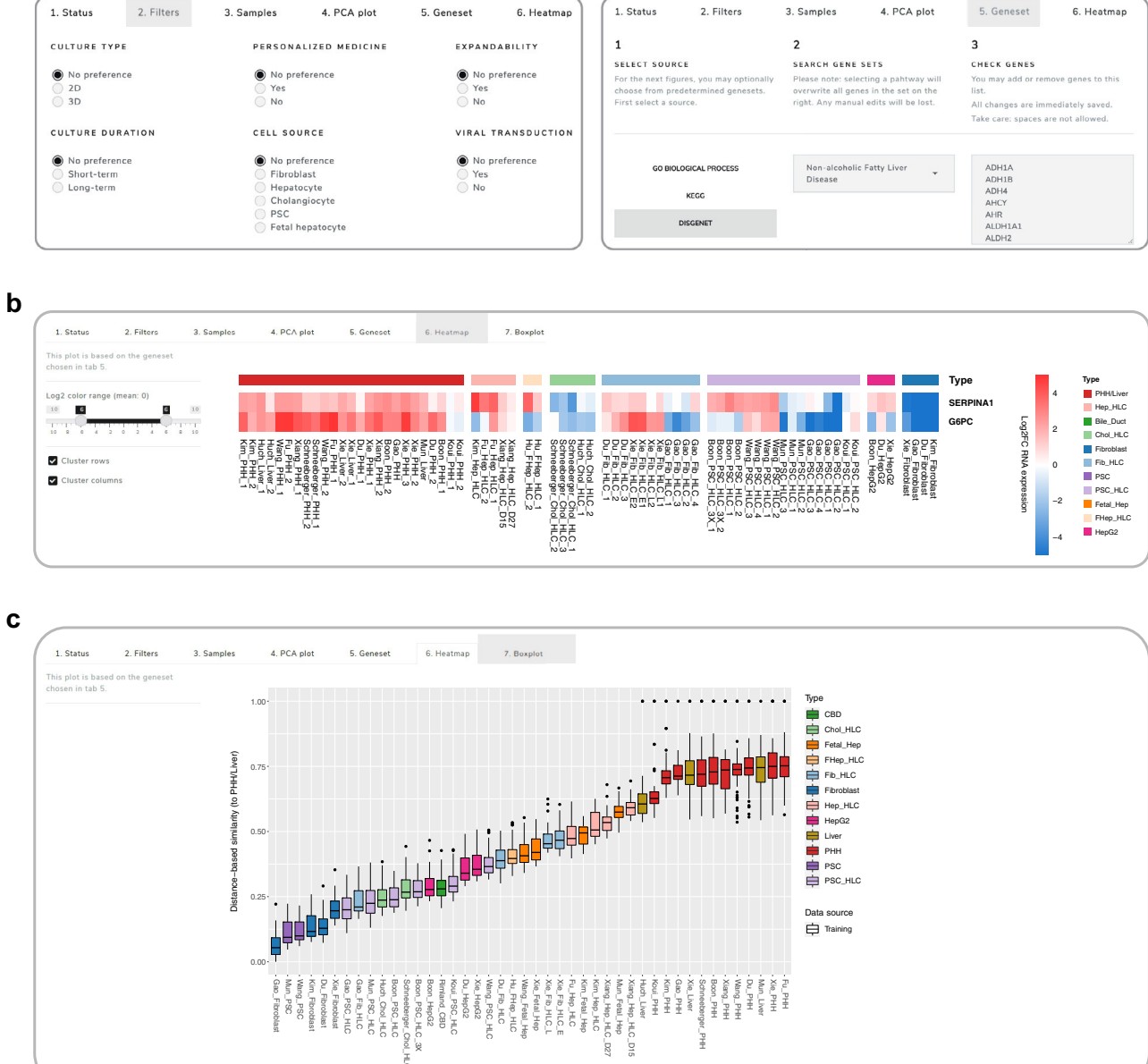

**Fig. 5 HLCompR web application for HLC selection. a** The HLCompR web application allows filtering for HLCs based on various properties and enables easy selection of gene sets relevant to hepatic functions and diseases. **b** Heatmap showing the relative expression of *SERPINA1* and *G6PC* between samples. **c** Boxplot showing DBS of non-alcoholic fatty liver disease-associated genes according to the DisGeNET database. Box-and-whisker plots are shown as median (line), interquartile range (box), and data range or 1.5x interquartile range (whisker).

question. We validated that our transcriptomic HLC comparison allows functional performance prediction in well-established models in our laboratory. The strong correlation between transcriptional and functional profiles supports the use of our transcriptomic comparison as a resource to select HLCs for specific research goals. Choosing the optimal HLCs is accommodated by our web application (HLCompR), which allows filtering by HLC properties and selecting custom gene sets. This functionality sets HLCompR apart from other comparison platforms such as CellNet, which are focused on evaluating the general cell identity of HLCs. The HLCompR web application also allows researchers to test HLCs generated in their own laboratories, including replications of HLC protocols included in this study or development of novel HLC protocols, if the PHH control is comparable to the training dataset. Therefore, we advise that protocols

describing new HLCs provide publicly available transcriptomic data alongside PHH controls that serve as a benchmark for cross-study comparability.

Besides transcriptional and functional similarity to PHHs, other characteristics may be important when selecting the optimal liver in vitro model, including the ability to derive HLCs from small biopsies for personalized medicine. This can be achieved by using skin-derived fibroblasts that are directly transdifferentiated (Xie, Gao, Du) or reprogrammed to induced-PSCs (iPSCs) that are subsequently differentiated to HLCs (Koui, Gao, Mun). Using a specific combination of TFs, direct transdifferentiation of fibroblasts resulted in HLCs that are transcriptionally closer to PHHs than most PSC-derived HLCs. Our approach specifies the effects of these TFs. The combination of TFs used by Xie, for example, not only induced activation of gene

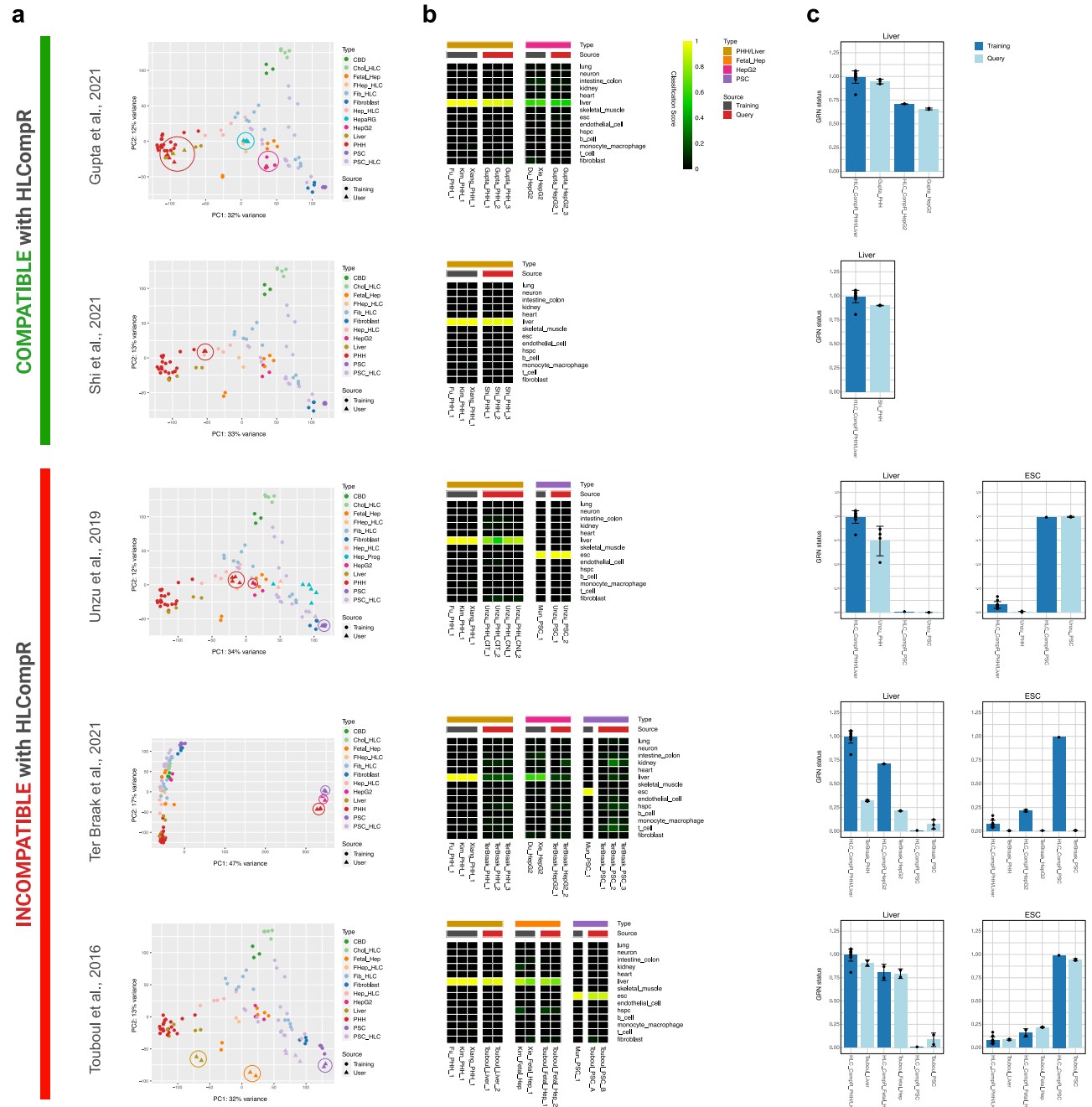

**Fig. 6 Addition of new HLC transcriptomes to the HLCompR web application. a** Principal component analysis on query datasets. Circles indicate the cell or tissue types in the query datasets that are also present in the training dataset (e.g., PHH, liver, HepG2, fetal hepatocyte, and PSC) or the HepaRG samples in Gupta[74] dataset. HLCompR compatibility is categorized based on the comparability of PHHs and liver tissue between the query and training datasets. **b** Cell/tissue classification heatmap of representative training samples and query samples. **c** Gene regulatory network status of liver and embryonic stem cells (ESCs) of all PHH, liver, HepG2, and PSC samples from training dataset and query dataset presented as mean ± standard deviation. Source data are provided in Supplementary Data 8.

networks regulated by the transduced TFs, but also of other networks (Fig. 3d). Addition of *ATF5* and *PROX1*, which were relatively poorly established in the Fib-HLCs of Xie may further improve hepatic differentiation.

Hepatocyte maturity cannot be solely defined by the transcriptome, but should also be determined by the proteome and ultimately the functionality. However, reliable comparison of hepatic functionality between HLCs requires complete standardization of methodology and PHH controls. In practice, this entails performing the assays in all HLCs simultaneously using

the same experimental setup. Given the difficulties in reproducing all HLCs, this would be highly challenging. Therefore, transcriptomic comparison may represent the most compelling solution to ensure standardization of multiple samples from various studies. Indeed, this allows evaluation of HLCs using a common benchmark of PHH controls from multiple studies, preventing over- or underestimation of hepatic characteristics caused by suboptimal PHH controls in individual studies. Additionally, transcriptomic comparison enables comprehensive characterization based on more than a handful of reported

marker genes. This includes assessment of multiple cell/tissue identities and the extent of hepatocyte maturation (Fig. 3).

In summary, this study provides a method and a web application (HLCompR) to compare and evaluate HLCs from multiple studies. We found that the transcriptome of hepatocyte-derived HLCs is currently most similar to primary hepatocytes. For personalized medicine purposes however, dermal fibroblasts are more readily available and improvements in direct transdifferentiation of fibroblasts have resulted in excellent HLCs. Importantly, our strategy allows identification of individual TFs or culturing conditions that might improve hepatic differentiation. Moreover, evaluation of HLCs alongside relevant control tissues provides insight into their tissue identity. These insights will guide improvement of HLC culture protocols, thereby advancing hepatic in vitro modeling and supporting regenerative strategies. Finally, although we focused on in vitro models of hepatocytes, the same method may be applied to evaluate in vitro models of other cell types or organs.

## Methods

**Data collection**. The PubMed database was systematically searched for studies that mention the development or evaluation of HLC culturing protocols. Search terms were selected to include studies performing long-term culturing of human primary hepatocytes or (trans)differentiation of human somatic or stem cells into HLCs, including functional or transcriptomic evaluation (Supplementary Fig. 1). The titles and/or abstracts of all hits were screened, including studies describing and evaluating new HLC protocols, and excluding studies using previously described HLC protocols. During subsequent full-text screening, only studies that functionally evaluated new HLC protocols alongside PHH as a common standard were selected for functional and expressional comparison in Fig. 1. Finally, from selected studies, only those providing publicly available bulk RNA-seq datasets with the inclusion of PHH/liver controls were included for analysis in our computational algorithm (Supplementary Fig. 1).

**Functional and expressional comparison of HLCs using reported data**. Quantification of functional assays and mRNA expression by qPCR was estimated based on the figures and graphs provided in the original studies. Functional and expressional data of HLCs were normalized to the corresponding PHH data from the same study and represented as a percentage. Pearson correlation between assays and/or gene expression was calculated and visualized using GraphPad Prism 8.

**Study approval and human subjects**. The study was approved by the responsible local ethics committees (Institutional Review Board of the University Medical Center Utrecht (STEM: 10-402/K; TcBio 14-008; Metabolic Biobank: 19–489), Erasmus MC Medical Ethical Committee (MEC-2014-060), and the Dutch Ethical Medical Council (Leiden University MC)). Tissue biopsies from livers of healthy donors were obtained during surgery in the Erasmus MC, Rotterdam. Human fetal livers were obtained from Leiden University Medical Centre (MC). All patient materials were used after written informed consent.

**Organoid establishment and culture**. Cholangiocyte-derived organoids were established and cultured as described previously[17]. To obtain the cells, liver biopsies were cut into small pieces and digested using 10 mg/ml Collagenase D (Sigma, 11088866001) for 20 min at 37 °C. The samples were then washed with cold Advanced DMEM/F12 (Gibco, 12634028) supplemented with 2 mM GlutaMAX (Gibco, 35050061), 10 mM HEPES (Gibco, 15630080), 100 U/ml PenStrep (Gibco, 15140122), and spun at 1500 rpm for 5 min. Cell pellet was plated in matrigel (Corning, 356231) and culture medium was added. Culture media was based on Advanced DMEM/F12 supplemented with 2 mM GlutaMAX, 10 mM HEPES, 100 U/ml PenStrep, 2% B27 without vitamin A (Gibco, 12587010), 10 mM Nicotinamide (Sigma, N0636), 1.25 mM N-Acetylcysteine (Sigma, A9165), 10% RSPO1 conditioned media (homemade), 10 nM Gastrin (Tocris, 3006/1), 50 ng/ml EGF (Peprotech, AF-100-15), 100 ng/ml FGF10 (Peprotech, 100-26), 25 ng/ml HGF (Peprotech, 100-39), 50 μg/ml Primocin (Invivogen, ant-pm-2), 5 μM A83-01 (Tocris, 2939/10), and 10 μM Forskolin (Tocris, 1099/10). For the first 3 days after isolation from biopsies, the medium was supplemented with 30% Wnt conditioned media (homemade), 25 ng/ml Noggin (Peprotech, 120-10C), and hES cell cloning recovery solution (Stemgent, 010014500). The medium was changed every 3–4 days and organoids were passaged 1:4–1:8 each week. Differentiation towards hepatocyte was initiated by culturing the organoids in culture medium supplemented with 25 ng/ml BMP7 (Peprotech, 120-03) for 5–7 days. The medium was then changed to Advanced DMEM/F12 supplemented with 2 mM GlutaMAX, 10 mM HEPES, 100 U/ml PenStrep, 2% B27 without vitamin A, 1.25 mM N-Acetylcysteine, 10 nM Gastrin, 50 ng/ml EGF, 25 ng/ml HGF, 100 ng/ml FGF19 (Peprotech, 100-32), 50 μg/ml

Primocin, 500 nM A83-01, 25 ng/ml BMP7, 10 μM DAPT (Sigma, D5942), and 30 μM Dexamethasone (Sigma, D4902) for 8 days.

Fetal hepatocyte-derived organoids were established and cultured as described previously[58]. Human fetal liver tissue was chopped and digested using 100 μg/ml Collagenase Type IV (Sigma, C5138) for 5 min. Cells were washed with Advanced DMEM/F12 supplemented with 2 mM GlutaMAX, 10 mM HEPES, 100 U/ml PenStrep, filtered through 100-μm filter, and plated in matrigel. After matrigel had solidified, HEP medium was added. HEP medium consisted of Advanced DMEM/F12 supplemented with 2 mM GlutaMAX, 10 mM HEPES, 100 U/ml PenStrep, 2% B27 without vitamin A, 15% RSPO1 conditioned media, 2.5 mM Nicotinamide, 1.25 mM N-Acetylcysteine, 3 μM CHIR-99021 (Sigma, SML1046), 50 μg/ml Primocin, 50 ng/ml FGF7 (Peprotech, AF-100-19), 50 ng/ml FGF10, 50 ng/ml HGF, 50 μM Y-27632 (Abmole Bioscience, M1817), 1 μM A83-01, 20 ng/ml TGFα (Peprotech, 100-16 A), 50 ng/ml EGF, and 10 nM Gastrin. Medium was refreshed every 2–3 days and organoids were passaged 1:2–1:5 every week.

**RNA sequencing of intrahepatic cholangiocyte- and fetal hepatocyte-derived organoids**. For RNA sequencing analysis, we included liver samples, intrahepatic cholangiocyte-derived organoids, and fetal hepatocyte organoids that were cultured as described above. RNA was isolated using Trizol LS reagent (Invitrogen) and stored at –80 °C until further processing. mRNA was isolated using Poly(A) Beads (NEXTflex). RNA integrity was assessed using the Agilent RNA 6000 Nano kit and concentrations were determined using the Qubit RNA HS Assay Kit. Only RNA samples with RIN > 8.0 were used for sequencing. Sequencing libraries were prepared using the Rapid Directional RNA-Seq Kit (NEXTflex) and sequenced on a NextSeq500 (Illumina) to produce 75 base long reads (Utrecht Sequencing Facility).

**Raw read processing and normalization pipeline**. Raw reads from the RNA-seq data were obtained from the European Nucleotide Archive (ENA, https://www.ebi.ac.uk/ena). Raw reads were processed using Galaxy (https://usegalaxy.eu/) web-based platform[79]. Sample quality was assessed using FastQC tool (Galaxy Version 0.72). Low quality reads and adapter sequences were trimmed using Cutadapt (Galaxy Version 1.66.6). Alignment of the raw reads and quantification of gene expression were performed using RNA STAR tool (Galaxy Version 2.7.2b). Reads were mapped to Gencode human reference genome sequence release 33 (GRCh38.p13) and Gencode comprehensive gene annotation v33, using default parameters. Read counts were obtained using the "–quantMode GeneCounts" option in the RNA STAR tool. Normalized counts were obtained by applying the DESeq2 variance-stabilizing transformation (VST) to the read counts using the 'DESeq2' R package[80] followed by quantile normalization using the 'preprocesscore' R package[81]. Additional information regarding our normalization pipeline is provided in Supplementary Note 1 and Supplementary Fig. 9.

**Principal component analysis**. Principal component analysis (PCA) was performed using normalized counts and plotted using the 'ggplot2' R package by adopting the R function 'plotPCA' (including top 5,000 highest variance genes) from the 'DESeq2' R package.

**Euclidean distance and distance-based similarity**. The Euclidean distance was calculated by applying the R function 'dist' (method=euclidean) to the normalized counts. Distance-based similarity score (DBS) was defined so that a DBS of '1' signifies perfect similarity to PHH controls and '0' signifies the sample least similar to PHH controls. The DBS for each sample was obtained using the following formula:

$$DBS = (Max_{PHH} - Dist_{PHH})/Max_{PHH} \qquad (1)$$

$Max_{PHH}$: the maximum distance value to PHH in the sample matrix of a certain gene set.

$Dist_{PHH}$: the distance value of a sample to PHH.

**Gene expression heatmaps**. To visualize gene expression, normalized counts were mean-centered per row or gene (Log2FC RNA Expression) and plotted in heatmaps using the 'pheatmap' R package. When cluster trees were absent in heatmaps, columns were ordered by the type of cell source.

**CellNet analysis**. The bulk RNA-Seq CellNet pipeline was employed to quantify gene expression estimates as previously described[65], using the 'cn_salmon' function for alignment to reference genome GRCh38. Classification and gene regulatory network (GRN) status analysis were performed using the 'cn_apply' function, based on the human cnProc_HS_RS_Jun_20_2017 object trained by 14 types of cells and tissues from 97 studies. Tissue classification scores were exported and plotted in heatmaps using the 'pheatmap' R package. GRN status scores were exported and plotted in bar graphs using the 'ggplot2' R package. Network influence scores of tissue-specific transcription factors were calculated using the 'cn_nis_all' function for 'liver'. Network influence scores were exported and plotted in heatmaps alongside the normalized expression of corresponding genes, using the 'pheatmap' R package.

**Classification of adult vs. fetal hepatocyte identity**. Top 5000 genes with the highest variance across different cell types (primary human hepatocytes, fetal hepatocytes, common bile duct, fibroblasts, and pluripotent stem cells) were used to build a Random Forest classifier using the "randomForest" R package. Samples used to train the classifier are listed in Supplementary Data 2. Performance of the classifier was evaluated using the out-of-bag error rate.

**Comparison of transcriptomic analysis to proteomes**. The gene expression of various HepG2 cells, relative to the control PHH/livers, was compared to the publicly available proteomic data of HepG2 cells and PHHs from Tascher et al.[68] Of the 3995 identified proteins, 3703 (93%) could be matched to a unique gene included in the transcriptomic analysis. Enrichment analysis was performed in ernichR[82] using differentially abundant proteins (Tukey $p$ value < 0.05, lfc > 1; data from Tascher et al.[68]) and differentially expressed genes (padj <0.05, lfc >1) as separate inputs. Proteome and transcriptome enrichment scores were defined as log2 of the odds ratio and log2 of the inverse odds ratio for upregulated and downregulated genes in HepG2 vs. PHH samples, respectively.

**Functional assays**. Urea secretion was measured using a urea assay kit (Abcam, ab83362) according to the manufacturer's instructions. Briefly, 1 mM ammonium chloride was added to the culture medium. After 24 h, urea concentration in the medium was measured. CYP3A4 activity was measured using a P450-Glo CYP3A4 assay kit (Promega, V9001) according to the manufacturer's recommendations. All data were normalized to total protein content measured using Pierce BCA Protein Assay Kit (Thermo Fisher Scientific, 23225).

**PHH/liver quality control for HLCompR inclusion**. Quality of new PHH/liver data was assessed using the Random Forest classifier. The RNA-seq data used in Fig. 2 served as a training dataset to build the classifier. A dataset is not recommended to be used when the new PHH/liver samples have PHH/liver classification probability below 45%.

**Statistics and reproducibility**. Statistical analyses (Pearson's and Spearman's correlation) were performed using GraphPad Prism 8. Results were considered significant when $p < 0.05$. Sample sizes are generally indicated in the figures. For transcriptomic studies we included a minimum of two samples per type of hepatocyte-like cell (HLC) model, if available. In experiments designed to compare the functional capabilities of HLCs, we included at least two biological replicates for each HLC model. Unless stated otherwise, bar graphs are shown as mean ± standard deviation and box-and-whisker plots are shown as median (line), interquartile range (box), and data range or 1.5x interquartile range (whisker).

**Reporting summary**. Further information on research design is available in the Nature Research Reporting Summary linked to this article.

## Data availability

Summary of previously published reported assays and gene expression data included in this study is available in Supplementary Data 1. List of RNA-sequencing data used in this study is available in Supplementary Data 2. Raw counts, normalized counts, and metadata of the RNA-sequencing data used in this study is available in Supplementary Data 4. Source data underlying the figures are available in Supplementary Data 3–8. Processed RNA-sequencing data generated in this study have been deposited in the NCBI Gene Expression Omnibus database under accession number GSE214097. Raw RNA-sequencing data are not publicly available due to potential information that could compromise donor consent. All materials supporting the findings of this study are available from the corresponding author upon reasonable request.

## Code availability

R code for running the HLCompR web application is available on GitHub (https://github.com/iardisasmita/HLCompR) and Zenodo (https://doi.org/10.5281/zenodo.7071219). Further requests can be directed to the corresponding author.

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

## Acknowledgements

The study was supported by ZonMW TAS ('Regenerating Intestinal Tissue with Stem cells' project) and Metakids ('Minilevertjes' project). The study was stimulated by the national multidisciplinary collaboration United for Metabolic Diseases (UMD) to improve care for patients with metabolic diseases.

## Author contributions

Conceptualization: A.I.A., I.F.S., M.M., and S.A.F.; formal analysis: A.I.A. and I.F.S.; investigation: A.I.A., I.F.S., I.P.J., and G.K.; resources: D.H., B.A., and S.A.F.; software: A.I.A., I.F.S., and G.K.; writing – original draft: A.I.A. and I.F.S.; writing – review & editing: A.I.A., I.F.S., I.P.J., M.M., E.E.S.N., and S.A.F.; visualization: A.I.A. and I.F.S.; supervision: E.E.S.N. and S.A.F; and funding acquisition: E.E.S.N. and S.A.F.

## Competing interests

The authors declare no competing interests.

## Additional information

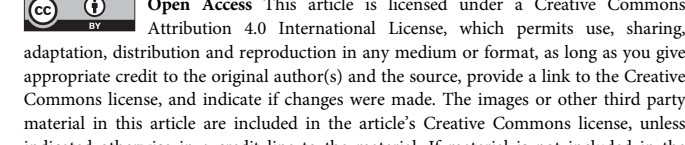

