## [Peer Review File · Communications Biology]

Reviewers' comments:

Reviewer #1 (Remarks to the Author):

Please find below my comments on the manuscript entitled "Comprehensive transcriptomic comparison of hepatocyte model systems" by Arif Ibrahim Ardisasmita and colleagues.

The manuscript describes a meta-analysis of transcriptomes of hepatocyte like cells (HLCs) generated in vitro from a diversity of source including human pluripotent stem cells and primary liver cells. These analyses reveal that HLCs from different origin display different functional properties and different transcriptomic profile. Some method especially HLCs derived from hepatocytes seem to generate cells with a "better" hepatic profile. They also show that HLCs can maintain the expression of genes related to their cell type of origin. Finally, they demonstrate that their approach can be used to define the interest of a specific HLCs for disease modelling. Finally, they have created HLCompR a new web interface to compare HLCs and to define their interest for specific application.

This is a solid and interesting study. It will be relevant for a broad number of labs working on HLCs and address a number of questions about their functionality. Thus, only few details need to be addressed:

Key marker missing in table 1 is AFP which is a foetal marker which is expressed in a majority of PSC-HLCs produced in vitro (but what about other HLCs?). It could be good to add this information.

Importantly, the authors should also consider the impact of culture conditions used by each lab to generate/grow HLCs. Indeed, these culture conditions are likely to change the transcriptomic profile of HLCs. This could interfere with their interpretation of Figure 3. This aspect should be discussed and probably use to dilute some of their conclusions.

Figure 3B suggests that foetal Heps, Hep-HLCs and PSC-HLCs all have a strong fibroblast signature. It is not clear if this analysis is meaningful. It could be very useful to give concrete example of genes actually express in HLCs and representative of each tissue. For example, what genes representing pluripotency are expressed in PSC-HLCs. It is likely that those genes are very generic and not truly specific.

Alternatively, the authors can't exclude the presence of contaminating cells in HLCs due to heterogenous differentiation. Again, they need to be more careful in their conclusion regarding this part.

They should describe clearly in the text specific criteria necessary for the use of HLCompR. This looks like a great tool and could be super useful. However, it should be usable by a broad number of groups. So, it would have also been incredibly useful to offer the possibility to add new data set directly to their web interface.

Reviewer #2 (Remarks to the Author):

Fuchs and colleagues conducted a comprehensive comparison study of the in vitro HLCs derived from various cell types and developed a computational algorithm allowing to compare whole transcriptome data across studies.

It is a in time study since different in vitro models for generating HLCs has been developed but no systematic comparison studies were performed. It is important to have this study as it will provide valuable information to the field especially when considering a specific model for a certain functional assay. It could also bring novel insights for optimizing the current HLC culturing/differentiation

protocols and serve as a platform for labs working on HLC system. From a biological perspective of view, the study is informative and solid.

There is only one concern about the in vitro models that been included. most of the in vitro HLC systems have been added in the study but not the hepatocyte organoid system reported in 2018(Hu et al., Cell). RNAseq data from primary hepatocytes and organoids has been published and available to add to the comparison. This is important as the community would like to know the gene expression signatures of the new model compared to the previous ones. detailed analysis and comparison of hepatocyte organoids and HPLs derived by other cell types regarding gene expression and function should be added. in any case, adding hepatocyte organoids to the study will make the study more complete unless it is technically not possible.

thank for the authors to performing this nice study and looking forward to have it expanded to more models and analysis.

Reviewer #3 (Remarks to the Author):

The authors provide a standardised and relatively unbiased strategy to globally and systematically compare the transcriptomes of different hepatocyte like cells/hepatocyte models to each other, which is important to consider when trying to identify the best or most suitable model for a particular research question or purpose. In addition, they take this one step further by linking their findings from the transcriptomic data to functional and proteomic data in order to highlight functional differences between different models and determine which model is best suited for a particular application or research question. This is all then made accessible and useful to the community via a dynamic web tool which allows researchers to filter and identify the most suitable model(s) based on various cell culture criteria and a gene set of interest.

Overall, the manuscript is well written and thorough, and I would like to thank the authors for providing such a comprehensive and extensive resource for hepatocyte research and hepatocyte models, which I think will be very useful for the liver community and for identifying suitable hepatocyte models for liver research. I have a few comments:

1). Given that zonation accounts for most of the gene expression variability in hepatocytes and regulates metabolic functions and properties of hepatocytes in vivo, I think it is quite important to discuss zonation throughout the paper in the context of the gene expression and functional assay data for the hepatocyte like cells/hepatocyte models. One example would be to highlight that the pathways found to be upregulated or down regulated in a given model are enriched or zoned periportally, midzonally, and/or pericentrally. Commenting on zonation in different parts of the paper as well as discussing zonation in a broader context and its relevance to hepatocyte like cells and hepatocyte models would be crucial.

2). From lines 188 to 197, where Chol-HLCs are discussed in the context of intestinal gene expression and identity, you can cite Aizarani et al. (2019) and highlight the findings showing gene expression differences between Chol-HLCs and primary cells at single-cell resolution, which include the expression of intestinal genes in Chol-HLCs (which are most relevant to the findings pertaining to this part). It would be good to include this to further backup this finding from other studies.

Regarding this part, it would also be important to discuss a bit why Chol-HLCs exhibit intestinal gene expression signature. For example, this could be due to the Chol-HLCs culture medium and conditions being very similar to those used for growing intestinal organoids.

Dear editor,

We would like to thank you and all reviewers for considering our work for publication in Communications Biology. We will address all remarks in a point-by-point manner in the following response:

Reviewer #1 (Remarks to the Author):

Please find below my comments on the manuscript entitled "Comprehensive transcriptomic comparison of hepatocyte model systems" by Arif Ibrahim Ardismita and colleagues.

The manuscript describes a meta-analysis of transcriptomes of hepatocyte like cells (HLCs) generated in vitro from a diversity of source including human pluripotent stem cells and primary liver cells. These analyses reveal that HLCs from different origin display different functional properties and different transcriptomic profile. Some method especially HLCs derived from hepatocytes seem to generate cells with a "better" hepatic profile. They also show that HLCs can maintain the expression of genes related to their cell type of origin. Finally, they demonstrate that their approach can be used to define the interest of a specific HLCs for disease modelling. Finally, they have created HLCompR a new web interface to compare HLCs and to define their interest for specific application.

1.0. This is a solid and interesting study. It will be relevant for a broad number of labs working on HLCs and address a number of questions about their functionality. Thus, only few details need to be addressed:

Author's Response: We thank the reviewer for the consideration of our work and are delighted that the aims and quality of this project were appreciated.

1.1. Key marker missing in table 1 is AFP which is a foetal marker which is expressed in a majority of PSC-HLCs produced in vitro (but what about other HLCs?). It could be good to add this information.

Author's Response: We agree that AFP is an important and interesting marker of hepatocyte immaturity and it would be nice to add it when considering hepatocyte maturity. In Fig. 1a, we rely on performed and reported expression analyses by the original papers. As shown in Supplementary Data 1, AFP expression was reported in 17 of the included studies. However, 6 out of 17 studies could not detect AFP expression in PHH. Since we expressed the value as a percentage of the PHH, we could only add the reported expression of AFP for 11 studies (Fig. 1a). As expected, all PSC-HLCs showed considerable expression of AFP compared to PHHs. This elevated AFP expression level was also observed in HLCs derived from fibroblasts and mesenchymal stem cells. In addition, we also described this immaturity in a more comprehensive manner (using more fetal markers and RNA-seq data) in Fig. 3c. Our results showed that all HLCs (regardless of the cell sources) displayed immature hepatic

expressional phenotype, in varying degrees.

1.2. Importantly, the authors should also consider the impact of culture conditions used by each lab to generate/grow HLCs. Indeed, these culture conditions are likely to change the transcriptomic profile of HLCs. This could interfere with their interpretation of Figure 3. This aspect should be discussed and probably use to dilute some of their conclusions.

Author's Response: We agree that the culture conditions used by each lab to generate HLCs from similar cell sources may change the transcriptomic profile of the fully established HLCs. We mentioned in the text that this effect exists, e.g. "Additionally, PSC-HLCs from Kouji also gained lung identity possibly due to their protocol involving differentiation towards multiple lineages (Fig. 3a & Fig. S2)." (line 200-201). To further strengthen this valid point, we now added another clarifying sentence to the preceding paragraph: "This heterogeneity in cell identity between HLCs starting from the same cell source (Wang-PSC-HLCs vs other PSC-HLCs and Gao-Fib-HLCs vs other fib-HLCs) reflects the effects of different culture protocols applied in each study." (line 195-197).

1.3. Figure 3B suggests that foetal Heps, Hep-HLCs and PSC-HLCs all have a strong fibroblast signature. It is not clear if this analysis is meaningful. It could be very useful to give concrete example of genes actually express in HLCs and representative of each tissue. For example, what genes representing pluripotency are expressed in PSC-HLCs? It is likely that those genes are very generic and not truly specific.

Author's Response: The way in which CellNet GRN status gene sets were established guarantees specificity of those gene sets for the 14 tissue/cell types included in the algorithm. Nevertheless, we agree that having some representative genes which make up the fibroblast and ESC GRN will provide more insights. Hence, we have added the gene expression heatmaps of specific fibroblast and ESC markers in Fig. S6.

1.4. Alternatively, the authors can't exclude the presence of contaminating cells in HLCs due to heterogenous differentiation. Again, they need to be more careful in their conclusion regarding this part.

Author's Response: Since this study investigates bulk RNA-Seq data, it determines the characteristics of the average cell in each HLC. Indeed, this bulk expression analysis cannot assess the degree of heterogeneous differentiation within HLC cultures. As such, the presence of unwanted cell identities could indeed be caused by a different (non-hepatic) cell population resulting from heterogeneous differentiation. To inform the reader about this limitation, we now added the following caveat to the results section of Fig. 3: "This gain of non-liver identity may occur in all cells or only in a subpopulation of cells, resulting from heterogeneous differentiation." (line 201-202).

1.5. They should describe clearly in the text specific criteria necessary for the use of HLCompR. This looks like a great tool and could be super useful. However, it should be usable by a broad number of groups. So, it would have also been incredibly useful to offer the possibility to add new data set directly to their web interface.

Author's Response: HLCompR does support the addition of new data directly to the web interface. However, for proper comparison, this relies on uniform mapping of FASTQ files according to the mapping used in this study and the establishment of the HLCompR training datasets. Therefore, we do provide a step-by-step guideline of how the mapping can be executed in the freely available galaxy platform in our github page, after which mapped data can be directly compared in the web interface. We notified readers of this possibility in the following sentence: "We therefore supported addition of new RNA-seq data to the HLCompR analysis using read counts processed according to our pipeline as inputs (<https://github.com/iardisasmitha/HLCompR>)." (line 310-311). If further explanation of this possibility is required, we are happy to do so.

Regarding the specific criteria stated in the text (originally line 325), we agree that in the previous version of the text, it was difficult to interpret which specific criteria were meant. Hence, we changed the sentence: "We recommend use of HLCompR only when specific criteria are met. Therefore, we included a quality check for new datasets, based on comparability of control samples (PHH/liver) using a Random Forest classifier." into: "We recommend use of HLCompR only when control PHH/liver samples are comparable to the training PHH/liver samples. We added a Random Forest classifier that automatically reports compatibility of a new dataset based on this parameter." (line 336-338)

As we show in Fig. 6, uniform mapping can still result in incompatibility with HLCompR. This may be caused by differences in library preparation protocols or sequencing machines (Supplementary Data 2). To optimize the chance of newly generated data being compatible with HLCompR, we compared the preparation protocols and now provide advice on how to perform bulk RNA-Sequencing to ensure compatibility in a separate paragraph "Since standardized mapping and RNA-seq processing may still result in incompatibility with HLCompR, we hypothesized that the type of RNA-seq library preparation or sequencer might influence compatibility (Supplementary Data 2). Based on this possibility, we suggest that libraries should be prepared using standard Illumina TruSeq RNA sample preparation kit and sequenced with Illumina sequencing machines. As we cannot guarantee compatibility only based on the methodological details, new datasets should always include PHH/liver control samples that should be comparable to the training PHH/liver samples." (line 341-346).

Reviewer #2 (Remarks to the Author):

Fuchs and colleagues conducted a comprehensive comparison study of the in vitro HLCs derived from various cell types and developed a computational algorithm allowing to compare whole transcriptome data across studies.

It is a in time study since different in vitro models for generating HLCs has been developed but no systematic comparison studies were performed. It is important to have this study as it will provide valuable information to the field especially when considering a specific model for a certain functional assay. It could also bring novel insights for optimizing the current HLC culturing/differentiation protocols and serve as a platform for labs working on HLC system. From a biological perspective of view, the study is informative and solid.

2.1. There is only one concern about the in vitro models that been included. most of the in vitro HLC systems have been added in the study but not the hepatocyte organoid system reported in 2018(Hu et al., Cell). RNAseq data from primary hepatocytes and organoids has been published and available to add to the comparison. This is important as the community would like to know the gene expression signatures of the new model compared to the previous ones. detailed analysis and comparison of hepatocyte organoids and HPLs derived by other cell types regarding gene expression and function should be added. in any case, adding hepatocyte organoids to the study will make the study more complete unless it is technically not possible.

thank for the authors to performing this nice study and looking forward to have it expanded to more models and analysis.

Author's Response: We thank the reviewer for considering and appreciating our work. With regards to the hepatocyte organoid model of Hu et al. (2018), we would like to note that this paper provided publicly available RNA-Seq data, but the library preparation was performed using a CEL-Seq2 protocol, which was incompatible with the other HLC data included in our study. We could include newly generated RNA-Seq data of fetal hepatocyte organoids cultured in our own lab according to the Hu et al. (2018) protocol, which was further detailed in Hendriks et al. (2021, Nature Protocols). Unfortunately, we were not able to generate long term adult hepatocyte-derived organoid cultures, and to the best of our knowledge, nor were other groups. As such, most of the follow-up work on the hepatocyte organoid paper (Hu et al. 2018) used fetal-derived hepatocyte organoids (Hendriks et al. 2021). Since we could not generate usable adult hepatocyte-derived organoids for RNA analyses or functional validation, we did not include them in our analysis.

Reviewer #3 (Remarks to the Author):

The authors provide a standardised and relatively unbiased strategy to globally and systematically compare the transcriptomes of different hepatocyte like cells/hepatocyte

models to each other, which is important to consider when trying to identify the best or most suitable model for a particular research question or purpose. In addition, they take this one step further by linking their findings from the transcriptomic data to functional and proteomic data in order to highlight functional differences between different models and determine which model is best suited for a particular application or research question. This is all then made accessible and useful to the community via a dynamic web tool which allows researchers to filter and identify the most suitable model(s) based on various cell culture criteria and a gene set of interest.

3.0. Overall, the manuscript is well written and thorough, and I would like to thank the authors for providing such a comprehensive and extensive resource for hepatocyte research and hepatocyte models, which I think will be very useful for the liver community and for identifying suitable hepatocyte models for liver research.

Author's Response: We are grateful for the constructive review and esteem of our work.

I have a few comments:

3.1. Given that zonation accounts for most of the gene expression variability in hepatocytes and regulates metabolic functions and properties of hepatocytes in vivo, I think it is quite important to discuss zonation throughout the paper in the context of the gene expression and functional assay data for the hepatocyte like cells/hepatocyte models. One example would be to highlight that the pathways found to be upregulated or down regulated in a given model are enriched or zoned periportally, midzonally, and/or pericentrally. Commenting on zonation in different parts of the paper as well as discussing zonation in a broader context and its relevance to hepatocyte like cells and hepatocyte models would be crucial.

Author's Response: This is a very interesting suggestion and we agree that addition of liver zonation transcriptomic analyses strengthens the paper. Therefore, we added an analysis of hepatocyte genes that are zoned either periportally or pericentrally in the human liver (Aizarani et al. 2019) in a new, separate paragraph in the Results section under subheading "Transcriptomic comparison reveals distinct liver-specific molecular signatures in HLCs" (Fig. 2g and Fig. S5): "This approach revealed that most HLCs expressed periportal and pericentral modules in a linear manner, indicating no strong zonation patterns. Deviating from this general pattern, the Chol-HLCs displayed a predominant periportal identity, which corresponded to relatively high expression of gluconeogenic genes. In contrast, HepG2 cells and the Fib-HLCs from Du exhibited a more pericentral identity (Fig. 2g), corresponding to higher expression of genes involved in cholesterol metabolism (Fig. 2f)," (line 150-160)

3.2. From lines 188 to 197, where Chol-HLCs are discussed in the context of intestinal

gene expression and identity, you can cite Aizarani et al. (2019) and highlight the findings showing gene expression differences between Chol-HLCs and primary cells at single-cell resolution, which include the expression of intestinal genes in Chol-HLCs (which are most relevant to the findings pertaining to this part). It would be good to include this to further backup this finding from other studies.

Author's Response: We thank the reviewer for the righteous suggestion to cite the finding of increased expression of intestinal genes (AGR2, MUC5AC, and MUC5B) in Chol-HLCs cultured in expansion medium by Aizarani et al. (2019). We have now referenced this finding in the following sentence: "This is in line with the finding of Aizarani et al. that intrahepatic cholangiocytes upregulate intestinal marker genes when cultured as organoids in Huch expansion medium." (line 209-211).

3.3. Regarding this part, it would also be important to discuss a bit why Chol-HLCs exhibit intestinal gene expression signature. For example, this could be due to the Chol-HLCs culture medium and conditions being very similar to those used for growing intestinal organoids.

Author's Response: We agree with the reviewer that the intestinal identity and underlying mechanism observed in Chol-HLCs is intriguing. In fact, we are currently preparing a new manuscript which explores exactly this topic. Therefore, we did not elaborate on this intestinal signature, which we believe deserves a more thorough assessment. This is outside of the scope of this paper, which focuses on the hepatic identity of the different HLC models.

REVIEWERS' COMMENTS:

Reviewer #1 (Remarks to the Author):

The authors have answered all the necessary comments.

Reviewer #2 (Remarks to the Author):

Thanks the authors to confirm the difficulties for longterm culture of hepatocyte organoids from adult liver. The authors need to mentioned about his in the discussion section that the broad community will receive the information.

Please add the fetal hepatocyte organoid data.

I have no other questions for this study. thanks the authors for the informative and solid work.

Reviewer #3 (Remarks to the Author):

I would like to thank the authors for addressing my comments and providing the zonation analysis. I have a few comments:

Regarding the zonation analysis the authors state:

156 This approach revealed that most HLCs expressed periportal and pericentral modules in a linear manner, indicating no strong zonation patterns.

Since this was a bulk analysis of zoned gene expression and does not provide single-cell resolution, one cannot phrase it as such and conclude that there are no strong zonation patterns. For example, it may be that in bulk the expression of both periportal and pericentral genes are well detected, but one cannot tell whether these genes are co-expressed in the same cells or whether the expression patterns for periportal and pericentral genes are non-overlapping across cells since this is a bulk analysis. The authors should just rephrase this statement and explain better and discuss the limitations of the bulk analysis in the context of zonation and the need for single-cell analysis to fully address zonation in HLCs. Nevertheless, as the authors highlight it's still interesting to see from the bulk analysis that certain HLCs have higher expression of periportal genes or pericentral genes.

To guide the eye, I recommend that the authors add legend a PP (periportal) or PC (pericentral) next to the corresponding metabolic pathways in the heatmap in figure 2f.

Dear editor,

We would like to thank you and all reviewers again for considering our work for publication in Communications Biology. We will address all remarks in a point-by-point manner in the following response:

Reviewer #1 (Remarks to the Author):

The authors have answered all the necessary comments.

Reviewer #2 (Remarks to the Author):

2.1. Thanks the authors to confirm the difficulties for longterm culture of hepatocyte organoids from adult liver. The authors need to mentioned about his in the discussion section that the broad community will receive the information.

Author's response: We have now mentioned our difficulties and reasons for not including the adult hepatocyte organoids in the main text "We did not include the adult hepatocyte-derived organoids because we were not able to culture them over long periods of time and, to the best of our knowledge, nor were other groups^{58,59}." (line 109-111)

2.2. Please add the fetal hepatocyte organoid data.

Author's response: We apologize for the unclear statement in our previous response. Already in the first version of this manuscript, we had included the fetal hepatocyte organoid data as Hu_FHep_HLCs and presented them in all RNA-seq analysis figures.

I have no other questions for this study. thanks the authors for the informative and solid work.

Reviewer #3 (Remarks to the Author):

I would like to thank the authors for addressing my comments and providing the zonation analysis. I have a few comments:

3.1. Regarding the zonation analysis the authors state:

156 This approach revealed that most HLCs expressed periportal and pericentral modules in a linear manner, indicating no strong zonation patterns.

Since this was a bulk analysis of zoned gene expression and does not provide single-cell resolution, one cannot phrase it as such and conclude that there are no strong zonation patterns. For example, it may be that in bulk the expression of both periportal and pericentral genes are well detected, but one cannot tell whether these genes are co-expressed in the same cells or whether the expression patterns for periportal and pericentral genes are non-overlapping across cells since this is a bulk analysis. The authors should just rephrase this statement and explain better and discuss the limitations of the bulk analysis in the context of zonation and the need for single-cell analysis to fully address zonation in HLCs. Nevertheless,

as the authors highlight it's still interesting to see from the bulk analysis that certain HLCs have higher expression of periportal genes or pericentral genes.

Author's response: We agree with and thank the reviewer for pointing this out. As such, we have made some adjustments to the text: "This approach revealed that most HLCs expressed periportal and pericentral modules in a linear manner (Fig. 2g). This may suggest that either there are no strong zonation patterns in most HLCs (assuming homogenous gene expression profiles in all cells) or there are zonation patterns that cannot be discerned due to the nature of bulk sequencing analysis. Single-cell RNA sequencing analysis is needed to fully address zonation in HLCs." (line 159-162)

3.2. To guide the eye, I recommend that the authors add legend a PP (periportal) or PC (pericentral) next to the corresponding metabolic pathways in the heatmap in figure 2f.

Author's response: Thanks to this recommendation, we have now added legends of liver zonation in Figure 2f.